# Simulation and Experiment of Gas-Solid Flow in a Safflower Sorting Device Based on the CFD-DEM Coupling Method

Zhizheng Hu 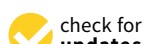, Haifeng Zeng *, Yun Ge, Wendong Wang and Jiangkun Wang

College of Mechanical and Electrical Engineering, Shihezi University, Xinjiang Uygur Autonomous Region, Shihezi 832003, China; huzhizheng@stu.shzu.edu.cn (Z.H.); gy_shz@163.com (Y.G.); wangwendong@stu.shzu.edu.cn (W.W.); wangjiangkun@stu.shzu.edu.cn (J.W.)
* Correspondence: zenghaifeng@shzu.edu.cn

**Abstract:** To study the movement characteristics and separation mechanism of safflower petals and their impurities under the action of airflow and lower the impurity rate in the cleaning operation process, integration of computational fluid dynamics (CFD) and discrete element method (DEM) codes was performed to study the motion and sorting behavior of impurity particles and safflower petals under different airflow inclination angles, dust removal angles and inlet airflow velocities by establishing a true particle model. In this model, the discrete particle phase was applied by the DEM software, and the continuum gas phase was described by the ANSYS Fluent software. The Box-Behnken experimental design with three factors and three levels was performed, and parameters such as inlet airflow velocity, airflow inclined angle, and dust remover angle were selected as independent variables that would influence the cleaning impurity rate and the cleaning loss rate. A mathematical model was established, and then the effects of various parameters and their interactions were analyzed. The test results show that the cleaning effect is best when the inlet airflow velocity is 7 m/s, the airflow inclined angle is 0°, and the dust remover angle is 25°. Confirmatory tests showed that the average cleaning impurity rate and cleaning loss rate were 0.69% and 2.75%, respectively, which dropped significantly compared with those from previous optimization. An experimental device was designed and set up; the experimental results were consistent with the simulation results, indicating that studying the physical behavior of safflower petals-impurity separation in the airflow field by using the DEM-CFD coupling method is reliable. This result provides a basis for follow-up studies of separation and cleaning devices for lightweight materials such as safflower petals.

**Keywords:** DEM-CFD coupling approach; airflow field; air classification; safflower; parameter optimization; simulation

## 1. Introduction

Safflower is a special cash crop that can be used as oil, medicine, feed, natural pigment and dye. It has strong adaptability, resistance to drought and cold and is easy to manage. Xinjiang has advantageous planting conditions and is the main area of safflower production in China. The annual planting area is more than 40,000 hm², and the yield is 3000~4000 t. In Xinjiang, safflower is mainly concentrated in the Tae basin (Tacheng, Yumin, Emin, Tuoli), which is known as "the hometown of Safflower in China". Yumin County alone has an annual planting area of approximately 11,300 hm². However, at present, the harvesting of safflower is not automated yet, we are still harvesting safflower artificially rather than mechanically, and there are problems that too many impurities mixed in the process of picking and drying safflower affect the subsequent intensive processing [1]. However, impurities and safflower silk have fewer confounding differences, and safflower silk is easily blown out with dust by air flow during operation using traditional grain cleaning machines, causing loss of safflower silk; at the same time, impurities cannot be effectively separated. Therefore, it is important to reduce losses during safflower cleaning and improve the sorting quality.

At present, the cleaning of safflower filaments after picking and drying is mainly carried out by vibrating screens. There are very few cleaning and sorting machines available, and the sorting efficiency is low. Generally, the cleaning devices used in agricultural production are air screen separation devices and air separation devices. An air screen separation device uses the combined action of air flow and a vibrating screen to separate materials, and an air separation device relies on air flow to separate materials with different physical properties [2]. Many scholars have studied the cleaning of rice and other agricultural materials. Rice cleaning is the operation of separating small debris from rice, such as rice straw, broken rice straw, husk and dust, by using the difference in physical properties among rice, rice straw and other components [3,4]. Shangpeng Ding studied the variation in particle motion trajectories in vertical and longitudinal fertilizer delivery tubes using the discrete element method [5]. Although air separation devices have been used in agriculture for several centuries, the research on the movement and separation behavior of safflower filaments in the flow field has been stagnant. The study of particle motion is mainly conducted using the DEM, and the study of the airflow field is conducted by the CFD.

The DEM is a numerical method for discrete media that is used to solve and analyze the motion law and mechanical characteristics of complex discrete systems [6]. Landry used a DEM to simulate the dynamic process of organic fertilizer particle (fertilizer compost) emissions from spiral solid fertilizer applicators [7]. Joseph et al. (2000) and Ketterhagen et al. simulated seed flow with different funnel geometries [8–10]. These results show that the DEM is an effective tool to simulate particle flow.

DEM is an interdisciplinary subject between mathematics, fluid mechanics and computer science. It analyses the system, including the physical phenomena of fluid flow, through numerical calculation and image display [11]. Due to a large amount of kinetic energy exchange in the process of safflower cleaning and the coupling effect formed by the fluid changes, the collision between particles and the interaction of fluid and particles makes the physical properties of the whole system very complicated. In the simulation study of safflower cleaning, if the simulation is conducted only with DEM or CFD, it is impossible to describe the interaction between airflow and the true safflower model, and the interaction between gas flow and irregular particles cannot be calculated correctly [12,13].

Therefore, we use the coupled method of discrete element and computational fluid dynamics. This is a new computational model that can characterize the real shape, physical parameters and particle collision motion of particles. It can also calculate the interaction force between fluid and particles through coupling with a fluent fluid. The motion distribution of particles can be clearly viewed through postprocessing, which is more comprehensive than the traditional simulation model [14]. Petit et al. used the CFD-DEM coupling method to study the particle movement behavior of air classifiers and improved the separation performance [15]. Dandan Han et al. used the DEM and CFD to simulate the gas-solid two-phase flow in a built-in blow-type corn precision metering device, studied the particle flow movement and its force, and improved and optimized the metering device [12]. All the above studies are based on the coupling method of discrete element and computational fluid dynamics, taking the air classifier and pneumatic seed metering device as the research objects, which indicates that the coupling method of discrete element and computational fluid dynamics is the development direction of the theoretical research of agricultural engineering in the future.

Therefore, this study takes the air separation device as the research object and adopts the discrete element computational fluid dynamics coupling method to simulate the particle cleaning process of safflower petals, bonded petals and stones. Combined with aerodynamics, the movement state and separation behavior of three kinds of particles in the air flow field were analyzed. The relationship between the inlet airflow velocity, the airflow inclined angle, the dust remover angle and the impurity rate and loss rate were studied, and the experimental results were optimized. In addition, an experimental device is designed to verify the reliability of the coupled method.

## 2. Materials and Methods

### 2.1. Structure and Working Principle of the Air-Separation Device

The safflower air-separation device consists of a control cabinet, blower, ventilation pipe, feed hopper, feed device, separating chamber, dust remover, and three receivers. There are also air inlets, three material outlets and baffles in the separation chamber. Its structure is shown in Figure 1.

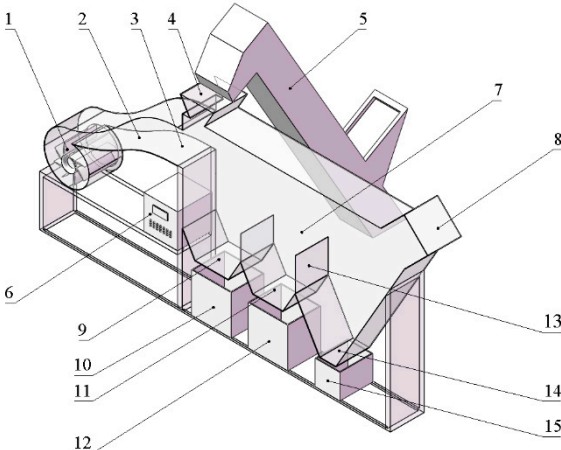

**Figure 1.** Schematic diagram of the structure of the safflower separation and cleaning device. 1. Blower; 2. Ventilation pipe; 3. Inlet (airflow); 4. Feed hopper; 5. Feed device; 6. Control cabinet; 7. Separating chamber; 8. Dust collector; 9. Outlet 1; 10. Stones receiver; 11. Outlet 2; 12. Bonded Safflower petals receiver; 13. Baffle; 14. Outlet 3; 15. Safflower petals receiver.

The working process of the device includes the feeding process, negative pressure air supply process and sorting process [16,17]. The sorting process is shown in Figure 2.

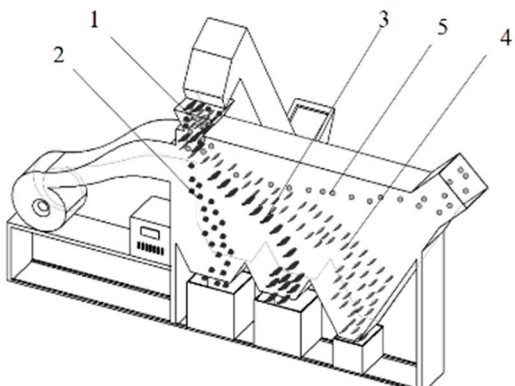

**Figure 2.** Working principle of the safflower separation and cleaning device. 1. Safflower impurities mixture; 2. Stones; 3. Bonded petals; 4. Safflower petals; 5. Dust.

Material is first transported through the feed device to the feed hopper and then into the sorting chamber. The airflow generated by the centrifugal fan enters the sort chamber, which is filled with airflow that can transport materials of different specific weights to different outlet ports, and under airflow, dust is also discharged from the dust remover.

### 2.2. Numerical Models

To analyze the characteristics of gas-solid two-phase flow, a mathematical model was established based on CFD and the DEM to theoretically explain the behavior of air flow and particles in separate cleaning devices. The material occupies a small proportion in the separation and cleaning device, and the gas phase is regarded as an incompressible fluid.

Safflower and its impurities are regarded as a collection of single particles, whose motion is governed by Newton's second law.

### 2.2.1. Gas Phase Mode

The two-way coupling method used in this paper considers the interaction between particles and fluid. For the two-way coupling method, transient state simulation should be performed because the effect of particles on the fluid is considered [18]. The continuity and momentum conservation equations of the continuous phase are given as follows:

$$\frac{\partial}{\partial t}\left(a_f\rho_f\right) + \frac{\partial}{\partial x_i}\left(a_f\rho_f u_j\right) = 0 \tag{1}$$

$$\frac{\partial}{\partial t}\left(a_f\rho_f u_i\right) + \frac{\partial}{\partial x_j}\left(a_f\rho_f u_i u_j\right) = -\frac{\partial\rho}{\partial x_i} + \frac{\partial}{\partial x_j}\left[a_f u_{eff}\left(\frac{\partial u_i}{\partial x_j} + \frac{\partial u_j}{\partial x_i}\right)\right] + a_f\rho_f g + F_s \tag{2}$$

where $F_s$ is the interaction term caused by the drag force between the particles and the fluid, and $a_f$ is the porosity near the particles, which can be calculated as follows:

$$a_f = 1 - \sum_{i=1}^{n}\frac{V_{p,i}}{V_{cell}} \tag{3}$$

where $V_{cell}$ is the volume of the selected CFD cell, $n$ represents the number of particles inside the cell, and $V_{p,i}$ is the volume of particles.

### 2.2.2. DEM Modeling

Computational particle mechanics models can describe interactions between particles and contact mechanics. Considering that the contact between particles and the particle velocity is changed based on the contact force, the softball dry contact model and the Hertz–Mindlin (no-slip) contact theory are used here in [19]. According to Newton's second law, the equation of motion of the $i$ particle is [20,21]:

$$m_i = \frac{dV_i}{dt} m_i g + P + \sum_{j=1}^{n_i}\left(F_{n,ij} + F_{t,ij}\right) \tag{4}$$

$$I_i = \frac{d\omega_i}{dt} = \sum_{j=1}^{n_i}\left(T_{t,ij} + T_{r,ij}\right) \tag{5}$$

where, $V_i$ is the velocity of particle $i$, m/s; $\omega_i$ is the angular velocity of particle $i$, rad/s; $I_i$ is the moment of inertia of particle $i$, kg·m$^2$; $m_i$ is the mass of particle $i$, kg; $g$ is the acceleration of gravity, m/s$^2$; $P$ is the force received when the particle moves relative to the airflow, N; $F_{n,ij}$ is the force normal component, N; $F_{t,ij}$ is the force tangential component, N; $T_{t,ij}$ is the tangential moment, N/m, and $T_{r,ij}$ is the rolling friction torque, N/m.

Any digital elevation model has two normal forces acting perpendicular to the contact plane and tangential forces acting on the contact plane [22,23]. Therefore, the total contact force acting on the particles on the contact plane can be analyzed in tangential and normal coordinates, which can be described by mathematical equations, as shown in Figure 3:

$$F = F_n^T + F_t^T \tag{6}$$

where $F$ is the total force acting on the particle, $F_n^T$ is the normal elastic–plastic contact force at the current time (T), and $F_t^T$ is the tangential elastic–plastic contact force at the current time (T).

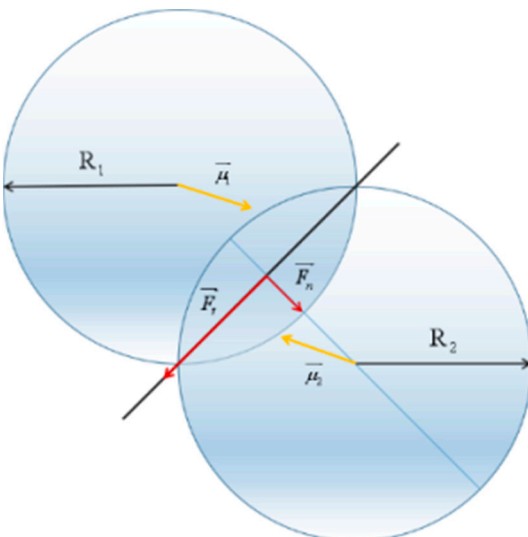

**Figure 3.** The collision force acting on a certain element of the spheres.

According to Hertz contact theory, the mathematical description of tangential and normal components is given by Equations (7) and (8):

$$F_{n,ij} = -\frac{4}{3}E^*\sqrt{R^*}(\delta_n)^{\frac{3}{2}}n_c - \sqrt{\frac{5}{6}k_n m^*}\frac{2\ln\varepsilon}{\sqrt{\ln^2\varepsilon + \pi^2}}(v_{n,ij}\cdot n_c)n_c \tag{7}$$

$$F_{t,ij} = -\frac{4}{3}G^*\sqrt{R^*\delta_n\delta_t} - \sqrt{\frac{5}{6}k_t m^*}\frac{2\ln\varepsilon}{\sqrt{\ln^2\varepsilon + \pi^2}}(v_{t,ij}\cdot n_c)n_c \tag{8}$$

where, $E^*$ is the equivalent elastic modulus, Pa; $R^*$ is the equivalent radius, m; $m^*$ is the equivalent mass, kg; $G^*$ is the equivalent shear modulus, Pa; $\delta_n$ is the normal overlap; $\delta_t$ is the tangential overlap; $k_n$ is the normal stiffness; $k_t$ is the tangential stiffness; $n_c$ is the unit vector connecting the centers of two particles; $\varepsilon$ is the elastic recovery coefficient; $v_{n,ij}$ is the relative normal velocity of particle j, m/s, and $v_{t,ij}$ is the relative tangential velocity of particle *i* to particle *j*, m/s.

The mathematical description of the tangential moment and rolling friction moment is shown in Equations (9) and (10):

$$T_{t,ij} = R_i F_{n,ij} \tag{9}$$

$$T_{r,ij} = -\mu R_i F_{t,ij}\omega_i \tag{10}$$

where $\mu$ is the coefficient of rolling friction; $R_i$ is the unit direction vector of particle i centroid to contact point, and $\omega_i$ is the angular velocity unit vector of the particle *i* contact point.

2.2.3. Forces and Torques from the Fluid to Particles

As shown in the Equation (4), there are four types of forces that the fluid exerts on particles. They are the drag force $F_d$, Saffman lift force $F_{lS}$, Magnus lift force $F_{lM}$ and buoyant force $F_b$, respectively. For the drag force of particles in a multi-particle system, the drag force model proposed by Felice (1994) is adopted [24]:

$$F_d = F_{d0}a_f^{-(y+1)} \tag{11}$$

where $F_{d0}$ represents the fluid drag force acting on a particle when there are no other particles, which is given by:

$$F_{d0} = \frac{1}{2}\rho_f C_D \frac{\pi d_p^2}{4} a_f^2 \left| v_f - v_p \right| \left( v_f - v_p \right) \tag{12}$$

where $v_f$ and $v_p$ are the fluid velocity and particle velocity, respectively. $d_p$ represents the particle diameter, and $C_D$ represents the fluid drag coefficient, which is given by the following:

$$\begin{cases} C_D = \frac{24}{Re_{p,a}} & Re_{p,a} \leq 1 \\ C_D = \left[ 0.63 + \frac{4.8}{Re_{p,a}^{0.5}} \right]^2 & Re_{p,a} \geq 1 \end{cases} \tag{13}$$

where $Re_{p,a}$ is the Reynolds number of the particle for the drag force. This Reynolds number is given by the following:

$$Re_{p,a} = \frac{\rho_f d_p a_f \left| v_f - v_p \right|}{\mu_f} \tag{14}$$

where $\mu_f$ is the viscosity of the fluid and the value of $\gamma$ in Equation (10) is given by:

$$\gamma = 3.7 - 0.65 exp \left[ -\frac{(1.5 - log_{10} Re_{p,a})^2}{2} \right] \tag{15}$$

The Saffman lift force $F_{lS}$ due to fluid shear motion is given by [25]:

$$F_{lS} = 1.615 d_p^2 \left( \frac{\rho_f \mu_f}{\left| \omega_f \right|} \right)^{\frac{1}{2}} C_{lS} \left[ \left( v_f - v_p \right) * \omega_f \right] \tag{16}$$

where $\omega_f$ is the fluid rotation velocity which is given by:

$$\omega_f = \nabla \times v_f \tag{17}$$

The lift force coefficient $C_{lS}$ for greater particle Reynolds numbers can be written as [26]:

$$C_{lS} = \begin{cases} \left( 1 - 0.3314 \beta^{\frac{1}{2}} \right) e^{Re_p/10} + 0.3314 \beta^{1/2} & Re_p \leq 40 \\ 0.5524 \left( \beta Re_p \right)^{1/2} & Re_p \geq 40 \end{cases} \tag{18}$$

where $Re_p$ is the Reynolds number of the particle for the Saffman lift force. This Reynolds number is given by:

$$Re_p = \frac{\rho_f d_p \left| v_f - v_p \right|}{\mu_f} \tag{19}$$

$\beta$ in Equation (18) is given by:

$$\beta = \frac{d_p \left| \omega_f \right|}{2 \left| v_f - v_p \right|} \tag{20}$$

The Magnus lift force $F_{lM}$ is given by the following [27]:

$$F_{lM} = \frac{\pi}{8}\rho_f d_p^3 \frac{Re_p}{Re_r} C_{iM} \left[ \omega \times \left( v_f - v_p \right) \right] \tag{21}$$

where the Magnus lift force coefficient $C_{lM}$ can be calculated, which allows an extension of this lift force to higher particle Reynolds numbers [28]:

$$C_{lM} = 0.45 + \left( \frac{Re_p}{Re_r} - 0.45 \right) e^{-0.5684 Re_r^{0.4} Re_p^{0.3}} \tag{22}$$

where $Re_r$ is the Reynolds number of particle rotation and is given by:

$$Re_r = \frac{\rho_f d_p^2}{\mu_f} \tag{23}$$

in which

$$\omega = 0.5 \nabla \times v_f - \omega_{p35} \tag{24}$$

The buoyant force on a particle $F_b$ can be calculated by the following:

$$F_b = -V_p \rho_f g \tag{25}$$

where $V_p$ is the volume of the particle.

Torque applied to the particles by the fluid $T_f$ in Equation (16) can be calculated as follows according to Rubinow and Keller (1961) [29]:

$$T_f = \frac{\rho_f}{2} \left( \frac{d_p}{2} \right)^5 C_R |\omega_r| w_r \tag{26}$$

where the coefficient for the rotation torque $C_R$ can be obtained from [29,30]:

$$C_R = \begin{cases} \frac{64}{Re_r} & Re_r \leq 32 \\ \frac{12.9}{Re_r^{1/2}} + \frac{128.4}{Re_r} & 32 < Re_r < 1000 \end{cases} \tag{27}$$

The above forces are substituted into Equation (4), and the torques into Equation (5). Then, the motions of the particles can be calculated.

2.2.4. Forces from Particles to Fluid

When particles first enter the flow field, the force acting on the flow field is obvious.

The forces exerted by the particles on the fluids $F_s$ in a CFD cell can be calculated as follows:

$$F_s = \frac{-\sum_{i=1}^{n} \left( F_d^i + F_{iS}^i + F_{iM}^i + F_b^i \right)}{V_{cell}} \tag{28}$$

where $V_{cell}$ is the volume of the cell and n represents the total number of particles in this cell. For the force exerted by the particles on the fluid, the basic principle is Newton's third law, namely, when the fluid exerts a force on the particle, the particle simultaneously exerts a force equal in magnitude and opposite in direction of the fluid.

*2.3. Simulation Model*

2.3.1. Simulation Geometry Model

To improve safflower purity, the safflower must be cleaned in the air-separation device after picking and drying is completed. SolidWorks software is used to create a model of the air-separating devices. The size of the air separating device model is shown in Figure 4. The model is imported into the ANSYS Workbench to divide the grid.

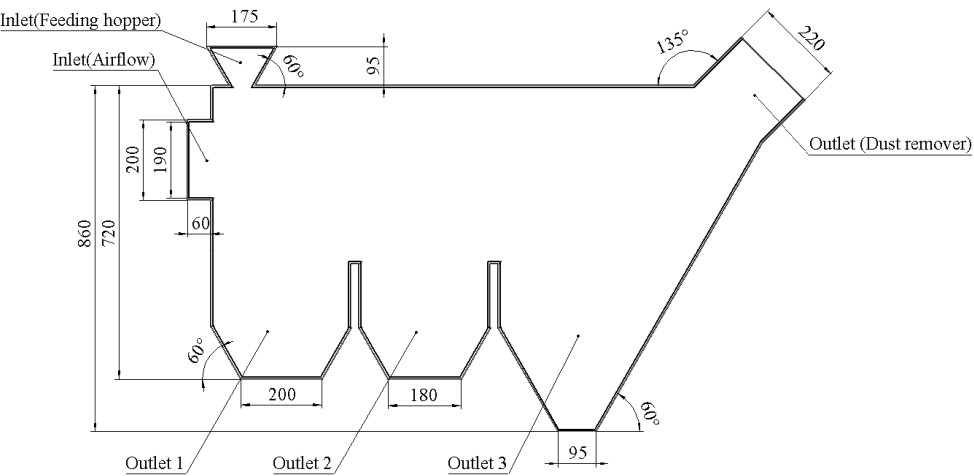

**Figure 4.** Dimensions of air-separating device unit.

2.3.2. Simulation Particles Model

Particles need to be modeled first when using digital elevation models. The interaction between particles is considered in the simulation process.

In this paper, for ease of calculation and reduction of calculation, only safflower petals, bonded petals and stones are considered screening objects, regardless of other impurities. The particle model is established according to the actual shape of the object and the particle is filled by the coordinate method. The filling sphere radius of the safflower petals is 0.25 mm and that of the bonded petals is 1 mm. The stones are filled with Tetrahedral Four, the default filling method of EDEM software, with a radius of 1 mm. The particle modeling is shown in Figure 5.

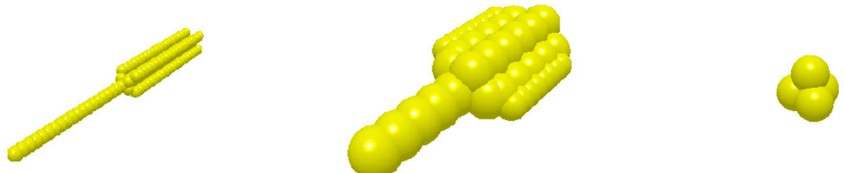

**Figure 5.** 3D models of particles.

Due to the different aerodynamic characteristics of safflower petals, bonded petals and stones, the vertically falling material exhibits different trajectories after being subjected to the horizontal airflow, and the material is acted upon by gravity G:

$$tana = \frac{P}{G} = \frac{k\rho A\left(v_q - v_w\right)^2}{mg} \qquad (29)$$

The resultant force of the three forces is F, as shown in Figure 6. When the horizontal airflow force P is constant, the smaller the gravity, the larger the moving direction angle α of the material particles. In aerodynamics, *tana* is the flight coefficient of the material in the flow field. As the physical properties such as particle size and density of the material are different, the flight coefficient in the same airflow is also different; when the airflow velocity is constant, the larger the flight coefficient is, the greater the horizontal displacement of particles driven by the airflow. Equation (29) shows that when the flight coefficient of the material is inversely proportional to its mass, stones with a larger mass sink obviously and fall into outlet 1, while safflower petals and bonded petals with relatively lighter quality are driven by the horizontal airflow force to make a horizontal projectile motion and fall into outlets 2 and 3.

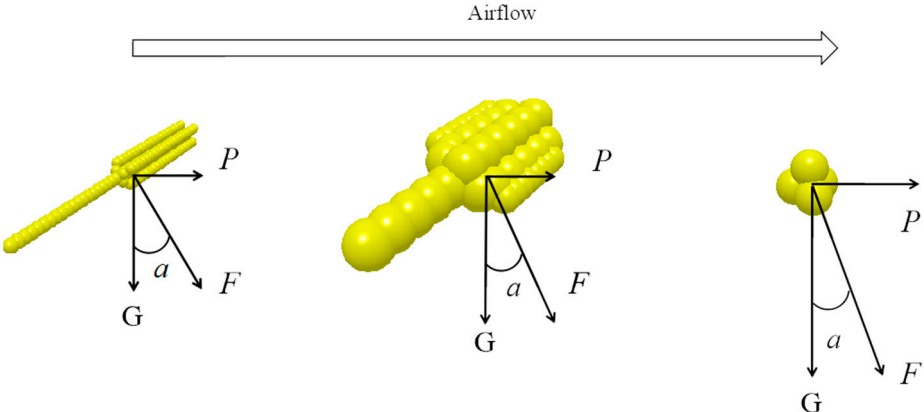

**Figure 6.** Force diagram of material particles.

The test stand was machined with methyl methacrylate polymer (acrylic). The values of some main parameters used in DEM-CFD coupling simulations are listed in Tables 1 and 2 [31].

**Table 1.** Variance analysis of regression model.

| Parameters | Filaments | Stones | Acrylic |
|---|---|---|---|
| Poisson's ratio | 0.3 | 0.18 | 0.4 |
| Shear modulus (Pa) | $9.4 \times 10^6$ | $1.8 \times 10^{10}$ | $3.5 \times 10^9$ |
| Density (kg/m$^3$) | 150 | 2650 | 1400 |

**Table 2.** Variance analysis of regression model.

| Parameters | Coefficient of Restitution | Coefficient of Static Friction | Coefficient of Rolling Friction |
|---|---|---|---|
| petals-petals | 0.1 | 0.2 | 0.15 |
| petals-bonded petals | 0.1 | 0.2 | 0.15 |
| petals-stones | 0.1 | 0.7 | 0.6 |
| petals-acrylic | 0.2 | 0.7 | 0.6 |
| bonded petals-bonded petals | 0.1 | 0.2 | 0.15 |
| bonded petals-stones | 0.2 | 0.7 | 0.6 |
| bonded petals-acrylic | 0.2 | 0.7 | 0.6 |
| stones-stones | 0.42 | 0.35 | 0.05 |
| stones-acrylic | 0.75 | 0.4 | 0.05 |

### 2.3.3. Setting of Simulation Parameters

Set the number ratio of safflower petals, bonded petals and stones to 12:1:1, and the safflower petals' production rate to 1200/s, the bonded petals' production rate to 100/s, and the stones' production rate to 100/s. The simulation time step is set to 33.081% of the Rayleigh time step, which is $1 \times 10^{-6}$ s, and the total simulation time is 5 s. The simulation in Fluent 2020R2 uses the standard k-ε turbulence model, and the time step is set to 100 times the time step in EDEM, which is $1 \times 10^{-4}$ s, and the convergence accuracy is $10^{-4}$. In the boundary condition setting, in all simulations, the velocity inlet and pressure outlet are adopted for the inlet and outlet of airflow. The gas phase motion is solved by the standard k-e turbulence model. We use standard wall functions as wall treatment methods.

### 2.3.4. DEM-CFD Coupling Simulation Method

In the DEM-CFD coupling simulation, CFD technique and particle motion was based on software of ANSYS Fluent 2020.0 and EDEM 2020. In ANSYS Fluent simulation, all differential governing equations were solved by applying finite volumes method and based on mass and momentum of the fluid phase (Equations (1) and (2)). First, the airflow

field was resolved by CFD solver. When a stable situation was obtained, gas field from CFD solver was transferred to DEM-CFD coupling interface which imported computation of forces acting on each particle. Then, the EDEM time step started at the end of fluid simulation time step. The calculated interface forces were delivered to the EDEM solver which computed the particle position, particle velocities and particle volume fraction until the end of CFD time step was reached. The next, DEM-CFD coupling interface took the particle translational and rotational motion data from the EDEM solver and computed the volume fractions and momentum exchange in the mesh cell of CFD. Finally, CFD solver used these data to solve the gas field for updating the fluid flow domain. The CFD and EDEM solvers entered into the cycles of the next time step until the airflow field again converged to a stable solution.

## 3. Results and Discussion

### 3.1. Mesh Independence Validation

For the transient numerical simulation, we need to analyze the relationship between the grid density and the calculation results under the same operating condition. In this simulation we take inlet airflow velocity of 8 m/s to study the outlet airflow velocity. Four different density grids were created to investigate the exit velocity at 5 s transient time. The validation parameters are shown in Table 3 and we have four grid types, A, B, C and D. Figure 7 is a comparison of velocities with different grid sizes. We have found that when the number of grids is greater than 25,996, the variation of calculation results is extremely small and tends to be stable. Therefore, we chose the model with 25,996 elements as our simulation model.

**Table 3.** Factors and levels of test.

| Type No. | Element Size (m) | Elements |
| --- | --- | --- |
| A | 0.06 | 7759 |
| B | 0.04 | 25,996 |
| C | 0.03 | 60,388 |
| D | 0.024 | 117,158 |

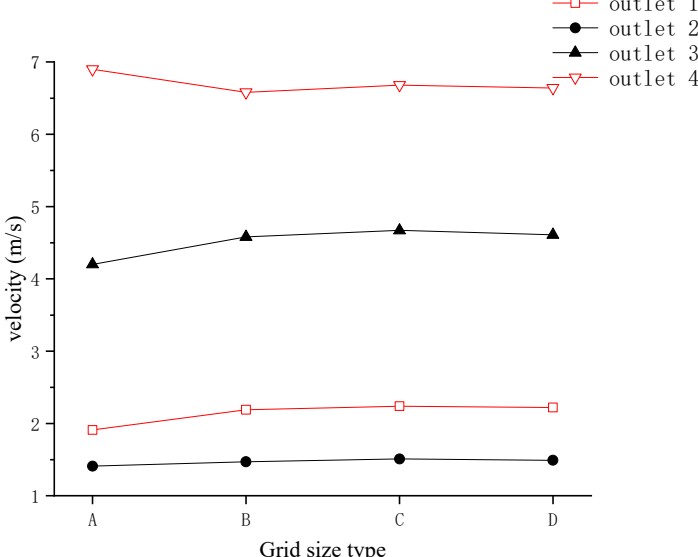

**Figure 7.** Exit velocity variation plots under different grid-size types.

### 3.2. Model Validation

To verify the reliability of using the DEM-CFD coupling method in the safflower petal cleaning simulation study, an experimental device is designed for the experimental verification of the stimulation of the safflower petal cleaning process. Figure 8 is the structure diagram of the experimental device. An anemometer is used to measure the inlet and outlet airflow velocities. The outlet airflow velocity at different velocities with a fixed airflow inclination angle and dust remover angle was tested. Simulation and experimental results of the inlet and outlet airflow velocities are presented in Figure 9. The simulation data are well-matched with the test data lists:

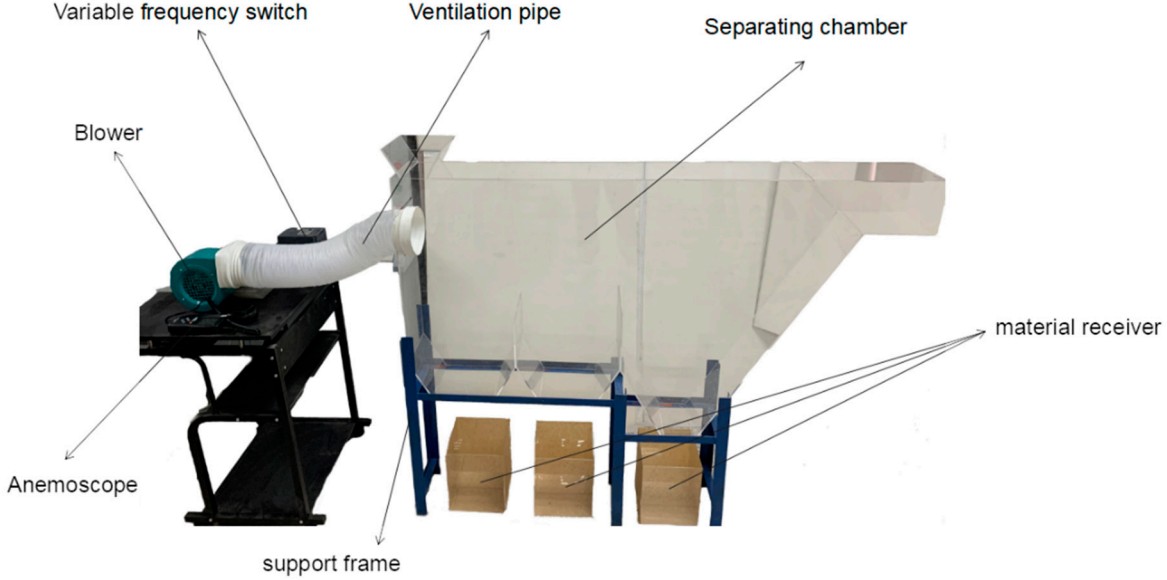

**Figure 8.** Experimental device of the physical diagram.

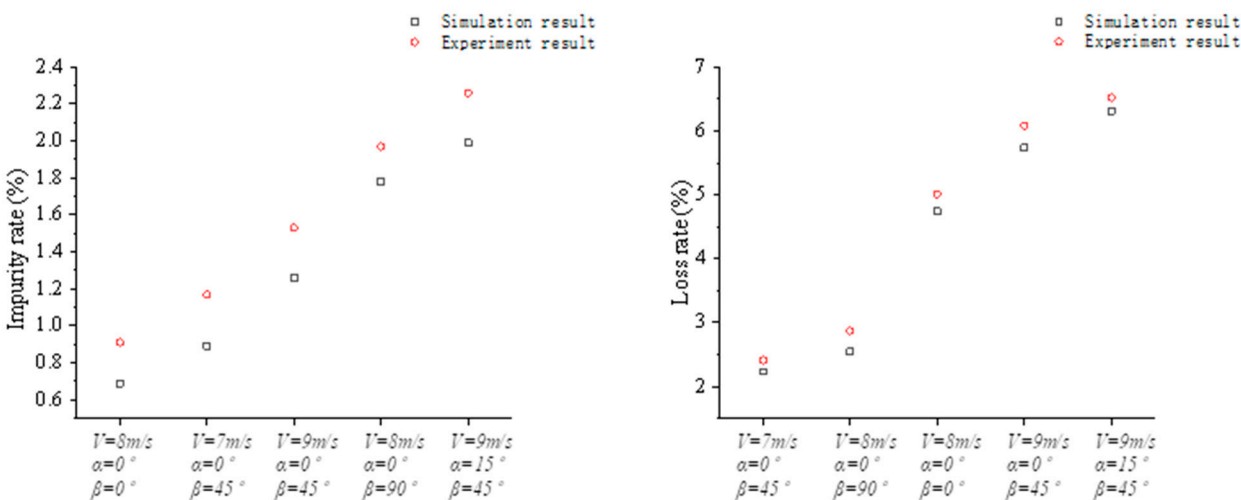

**Figure 9.** Comparison diagram of cleaning efficiency between simulation and experiment.

Under different parameters, the changing trends of impurity content and loss rate in the cleaning process of simulation and experiment are shown in Figure 10. It can be seen from the figure that after changing the experimental parameters, the change tendencies of the experimental results and the simulation results are basically consistent. However, it is also found that there are some errors in these data. This is due to the randomness of the volume and mass of safflower petals and bonded petals in the experiment compared with that in the simulation. In general, the experimental data are close to the simulation

data, which verifies the accuracy and feasibility of the simulation research on safflower petal cleaning based on the DEM-CFD coupling method.

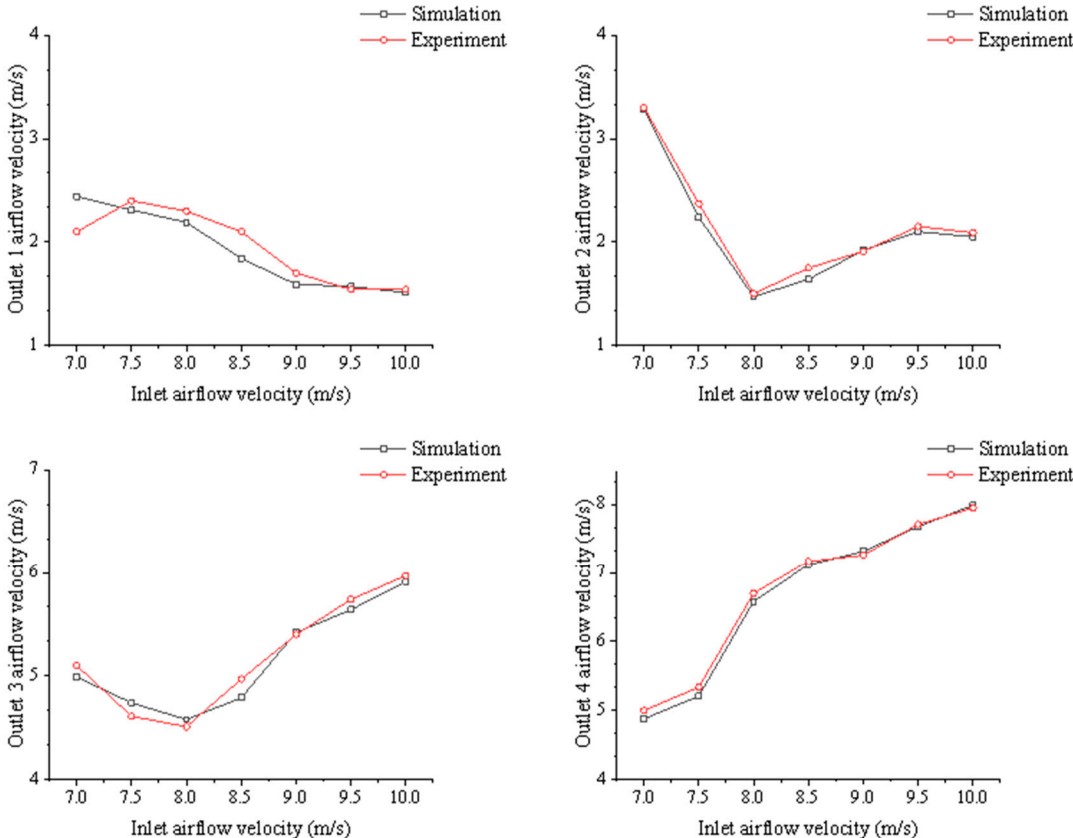

**Figure 10.** Comparison between experiment and simulation results.

### 3.3. Analysis of Single-Factor Simulations' Results

3.3.1. Influence of Airflow Inclined Angle on Mixture Screening

We set the inlet airflow velocity to 8 m/s and the angle of the dust remover to 45°. The changes in the airflow field and particle motion trace at different airflow inclination angles are shown in Figures 11–13. For the convenience of observation, the real particle shape is simplified as a sphere in Ensight2020R2 postprocessing, the safflower petals are scaled by 0.01, and the bonded petals and stones are scaled by 0.02. According to the analysis of Figure 11, laminar flow exists in the airflow under three conditions, and turbulence appears at the lower outlet, resulting in an unstable flow field. The airflow velocity near the wall is relatively small, and the airflow velocity at the air inlet is higher than that at other outlets. There is a transition zone of gas flow velocity in the sorting chamber. As the mixture particles fall from the feed hopper, the gas flow interacts with the particles, is resisted by the particles, and then spreads around, resulting in a certain loss of gas flow velocity.

According to the analysis of Figure 13, the local behavior of particle movement is helpful to understand the particle movement in the sorting chamber. When the material falls freely to a certain position after entering the sorting chamber from the feed hopper, it starts to do horizontal projectile movement under the action of air flow. With increasing airflow inclination angle, the particle velocity of the safflower petals increases obviously at the beginning of the middle stroke, and the number of high-speed safflower petal particles increases. With the increase in the axial airflow force on particles, the resultant force of particles in the airflow field becomes larger, the flying distance and flying height of safflower petals increase, the speed of bonded petals and stones increases, more bonded petals and stones are entrained by the flight of the safflower petals, and some petals and

stones also fly to outlet 3, which finally leads to an increase in the impurity rate. Due to the increase in the airflow inclination angle, some safflower petals are brought to the dust remover under the action of the airflow field, which leads to an increase in the loss rate. By comparing Figures 12 and 13, it can be seen that the airflow inclined angle has little effect on the flow field but has a great influence on the particle movement and separation effect.

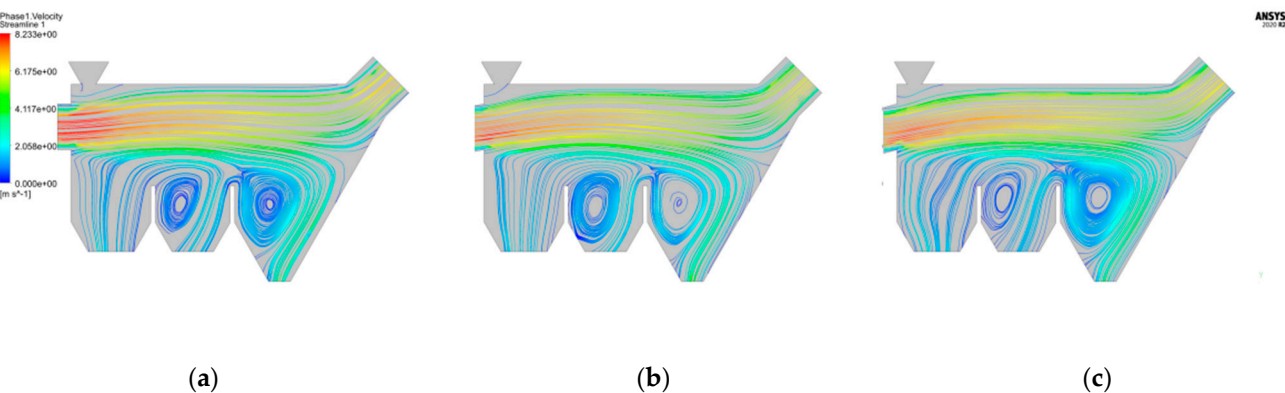

(**a**)                                      (**b**)                                      (**c**)

**Figure 11.** Streamline diagram of different types of airflow inclined angles: (**a**) 0°; (**b**) 7.5°; (**c**) 15°.

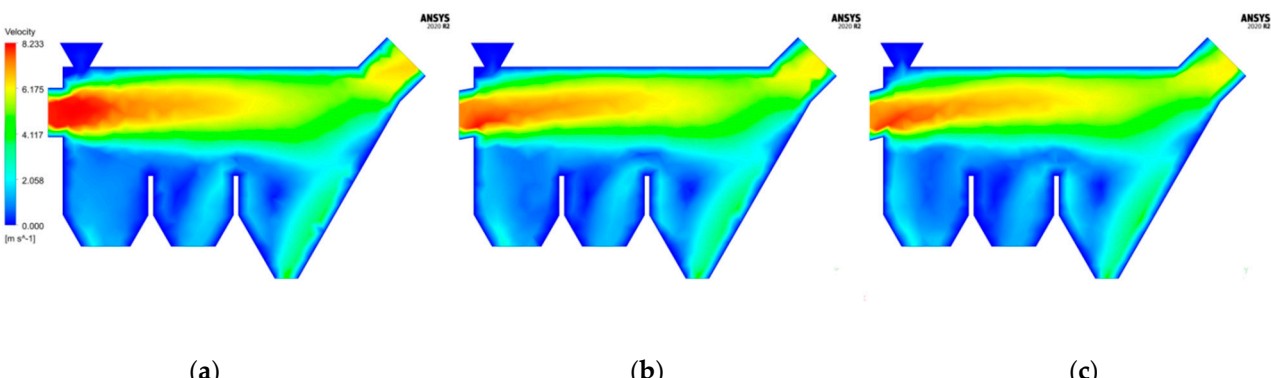

(**a**)                                      (**b**)                                      (**c**)

**Figure 12.** Airflow velocity contour plots of different types of airflow inclined angles: (**a**) 0°; (**b**) 7.5°; (**c**) 15°.

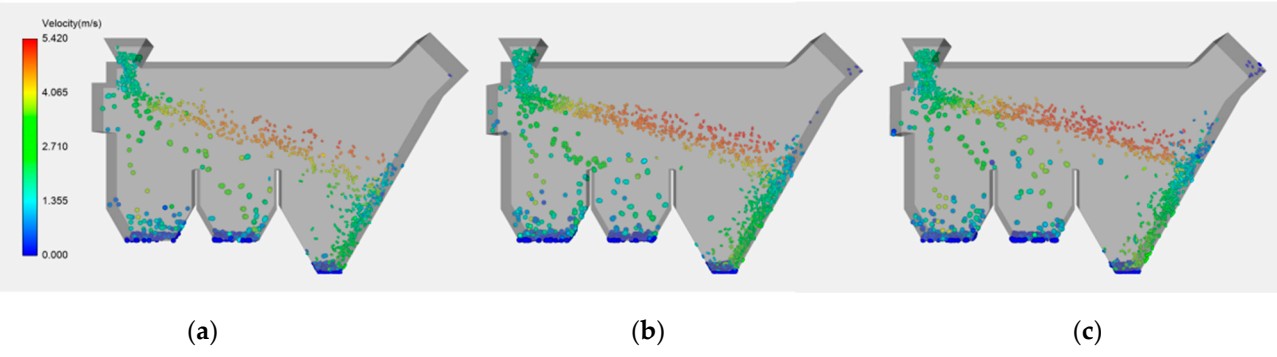

(**a**)                                      (**b**)                                      (**c**)

**Figure 13.** Particle motion of different types of airflow inclined angles: (**a**) 0°; (**b**) 7.5°; (**c**) 15°.

Figure 14 shows that the axial average velocity of each component of the mixture increases with increasing airflow inclination angle. The axial average velocity of the safflower petals is greater than that of the bonded petals and stones, and the axial speed of safflower petals increases obviously with increasing airflow inclination angle.

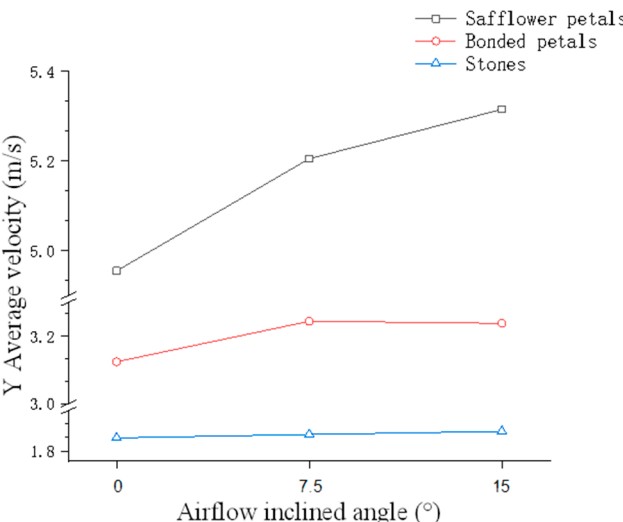

**Figure 14.** The influence of the airflow inclined angle on the axial average velocity of the mixture components.

### 3.3.2. Influence of Inlet Airflow Velocity on Mixture Screening

We set the airflow inclined angle to 0° and the angle of the dust remover to 45°. The changes in the airflow field and particle motion trace at different inlet airflow velocities are shown in Figures 15 and 16. According to the analysis of Figure 15, the change in inlet airflow velocity only has an effect on the flow rate of the airflow field and has no obvious effect on the change in airflow motion trend.

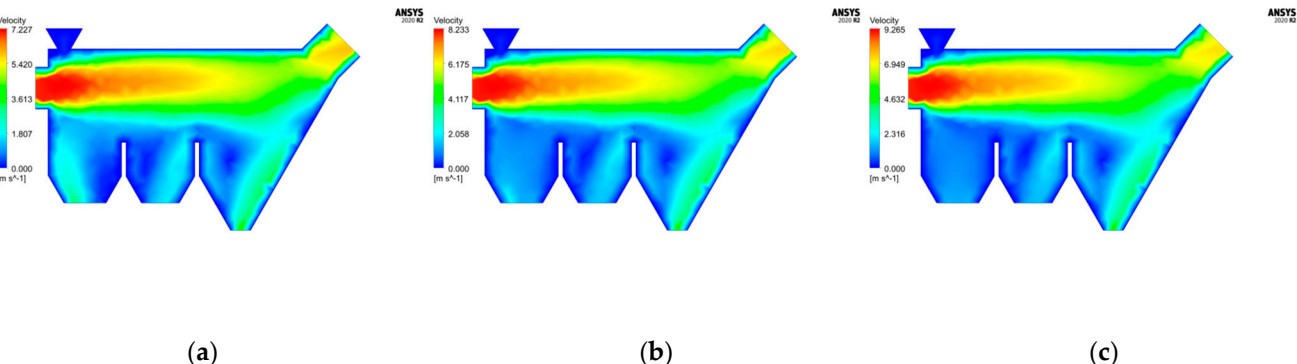

(**a**)  (**b**)  (**c**)

**Figure 15.** Airflow velocity contour plots of different inlet airflow velocity: (**a**) 7 m/s; (**b**) 8 m/s; (**c**) 9 m/s.

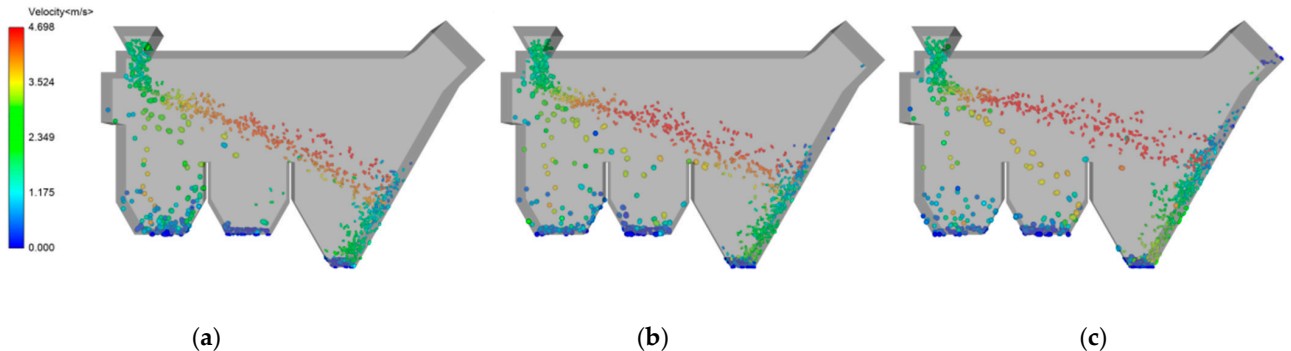

(**a**)  (**b**)  (**c**)

**Figure 16.** Particle motion of different inlet airflow velocity: (**a**) 7 m/s; (**b**) 8 m/s; (**c**) 9 m/s.

According to the analysis of Figure 16, when the inlet airflow speed is adjusted to 7 m/s, the wind power is weak. Due to the large specific gravity of stones and filaments, under the action of airflow, the resultant force is downward, and most of them settle at outlet 1. Very small amounts of petals and stones are brought to outlet 3, and the impurity rate is low. Under weak wind, the speed of the filament is lower, and part of the filament falls at outlet 2, resulting in an increase in the loss rate. The increase in the inlet airflow velocity leads to an increase in the dust remover velocity, which causes part of the safflower petals to move to the dust remover under the action of the airflow. It was found from the observation of the particle motion trajectory that the bonded petals and the stones began to gather towards outlet 2 and outlet 3 when the inlet airflow velocity increased, which led to an increase in the impurity rate. By comparing Figures 15 and 16, the airflow inclined angle has little effect on the flow field but has a great influence on the particle movement and separation effect.

Figure 17 shows that the axial average velocity of each component of the mixture increases with increasing airflow inclination angle. The axial average velocity of the safflower petals is greater than that of the bonded petals and stones, and the axial speed of safflower petals increases obviously with increasing inlet airflow velocity.

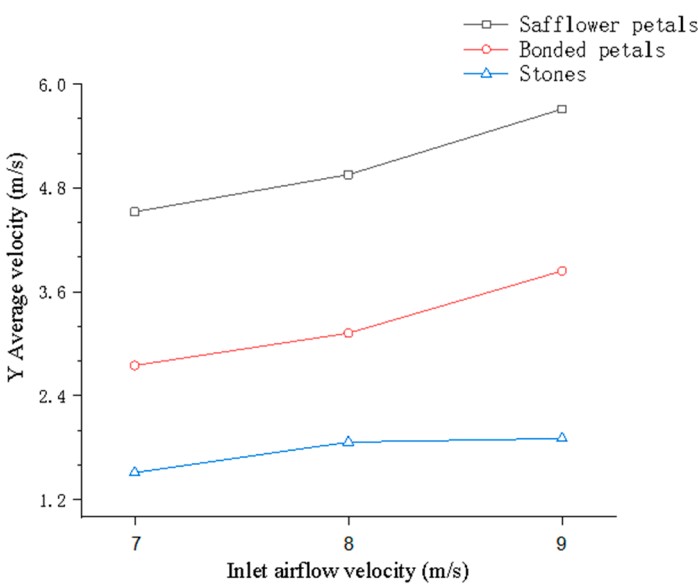

**Figure 17.** Influence of inlet airflow velocity on the axial average velocity of the mixture components.

### 3.3.3. Influence of Dust Remover Angle on Mixture Screening

We set the inlet airflow velocity to 8 m/s and the airflow inclined angle to 0°. The changes in the airflow field and particle motion trace at different angles of the dust remover are shown in Figures 18 and 19. Through the comparison with Figure 20, it is found that the angle of the dust remover has a greater effect on the airflow field. When the angle of the dust remover is at 0° and 90°, the airflow velocity loss is serious, and there is turbulence on the upper wall surface. Moreover, the airflow field is more unstable. Additionally, when the angle of the dust removal port is 0° and 90°, the velocities at outlet 2 and outlet 3 are higher than in the case of the angle of the dust removal port of 45°, and the velocity of the airflow in the near-wall region is relatively smaller. There is a transition zone in the separation chamber for the air flow velocity. When the angle of the dust remover is 45°, the transition zone is excessively stable, but when the angle of the dust remover is 0° and 90°, the transition zone of the air flow deviates. When the mixture falls from the feeding hopper, the gas flow interacts with the particles, and the gas flow is resisted by the particles and is then diffused around, resulting in a loss of gas flow velocity. It was found that the loss of airflow velocity is more obvious when the dust remover angle is 0° and 90°.

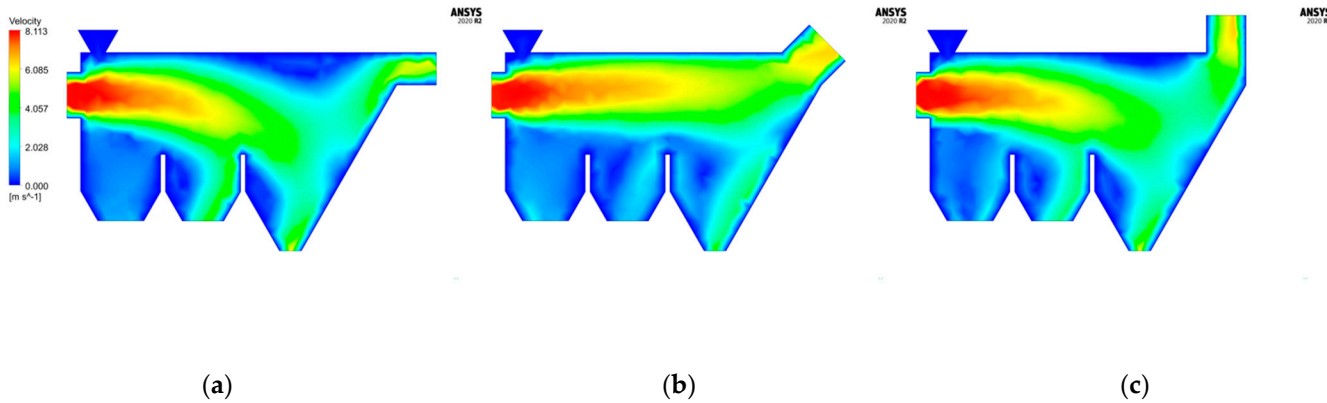

**Figure 18.** Airflow velocity contour plots of different types of dust remover angles: (**a**) 0°; (**b**) 45°; (**c**) 90°.

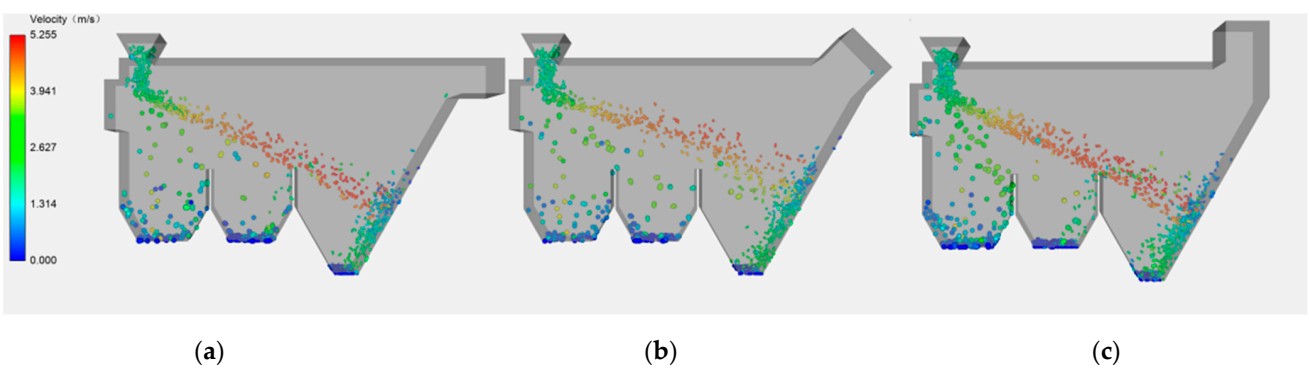

**Figure 19.** Particle motion of different types of dust remover angles: (**a**) 0°; (**b**) 45°; (**c**) 90°.

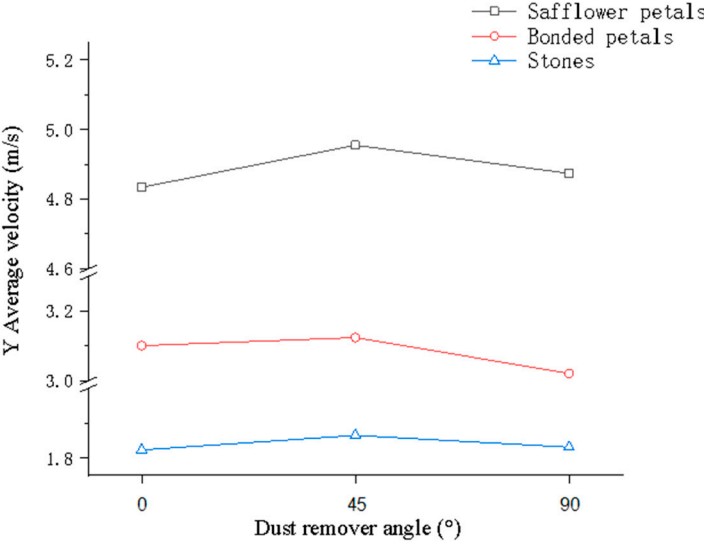

**Figure 20.** Influence of dust remover angle on the axial average velocity of the mixture components.

According to the analysis of Figure 20, when the angle of the dust remover is 0° and 90°, the airflow field is unstable, which causes a portion of the safflower petals to fall into outlet 2, resulting in an increase in the loss rate. The distance between particles is small, and there are more high-speed particles. Some particles are impacted on the top of the baffle plate between outlets 2 and 3 due to the airflow of the unstable airflow field, which affects the movement of particles and the separation and cleaning effect. When the angle of the dust remover is 45°, compared with the other two angles, the particles are relatively

dispersed, the particle spacing is larger, and the safflower petal particles stably fall into outlet 3. By comparing Figures 18 and 19, it is found that the dust remover angle has a great influence on the airflow field and particle movement, thus affecting the sorting efficiency.

### 3.4. Simulation Trial Design and Analysis

3.4.1. Response Surface Method Test Scheme

Single-factor simulation results and analysis were combined using the three-factor three-level Box–Behnken experimental design principle, with the inlet airflow velocity X1, airflow inclination angle X2 and dust remover angle X3 as independent variables, and the impurity rate Y1 and loss rate Y2 as response values. The test factors and levels are shown in Table 4.

**Table 4.** Factors and levels of tests.

| Code | $X_1/(m \cdot s^{-1})$ | $X_2/(°)$ | $X_3/(°)$ |
|------|------|------|------|
| −1 | 7 | 0 | 0 |
| 0 | 8 | 7.5 | 45 |
| 1 | 9 | 15 | 90 |

3.4.2. Regression Equation and Significance Analysis

The test plan and test results are shown in Table 5.

**Table 5.** Factors and levels of tests.

| No. | $X_1$ | $X_2$ | $X_3$ | $Y_1/\%$ | $Y_2/\%$ |
|------|------|------|------|------|------|
| 1 | 1 | 0 | 1 | 1.18 | 4.85 |
| 2 | 0 | 0 | 0 | 2.05 | 2.03 |
| 3 | 1 | 1 | 0 | 1.99 | 6.31 |
| 4 | −1 | 0 | −1 | 0.83 | 3.95 |
| 5 | 0 | 0 | 0 | 1.93 | 2.04 |
| 6 | 0 | 0 | 0 | 2.14 | 2.19 |
| 7 | 0 | 1 | 1 | 0.74 | 5.63 |
| 8 | −1 | −1 | 0 | 0.89 | 2.23 |
| 9 | 0 | 0 | 0 | 2.13 | 2.24 |
| 10 | −1 | 0 | 1 | 0.84 | 4.31 |
| 11 | 1 | 0 | −1 | 1.68 | 7.34 |
| 12 | 1 | −1 | 0 | 1.26 | 5.74 |
| 13 | 0 | −1 | −1 | 0.69 | 4.74 |
| 14 | 0 | −1 | 1 | 1.78 | 2.54 |
| 15 | 0 | 1 | −1 | 1.54 | 5.25 |
| 16 | 0 | 0 | 0 | 2.58 | 2.64 |
| 17 | −1 | 1 | 0 | 0.91 | 4.11 |

With the help of the Design-Expert 11 software, analysis of variance of the regression model for the impurity rate $Y_1$ and the cleaning loss rate $Y_2$ was performed, as shown in Table 6. The quadratic regression models of $Y_1$ and $Y_2$ are obtained as follows:

$$Y_1 = -32.242 + 7.948X_1 - 0.003867X_2 + 0.057244X_3 + 0.023667X_1X_2 - 0.002833X_1X_3 - 0.0014X_2X_3 - 0.47925X_1^2 - 0.007542X_2^2 - 0.000274X_3^2 \tag{30}$$

$$Y_2 = 81.10650 - 21.291X_1 + 0.124567X_2 + 0.038539X_3 - 0.043667X_1X_2 - 0.015833X_1X_3 + 0.001911X_2X_3 + 1.471X_1^2 + 0.015973X_2^2 - 0.000698X_3^2 \tag{31}$$

**Table 6.** Variance analysis of regression model.

| Source of Variance | Impurity Rate ($P_h$)/% | | | | Loss Rate ($G_h$)/% | | | |
|---|---|---|---|---|---|---|---|---|
| | Sum of Squares | DOF | F | Significance Level P | Sum of Squares | DOF | F | Significance Level p |
| Model | 5.37 | 9 | 9.54 | 0.0035 | 45.51 | 9 | 43.21 | <0.0001 ** |
| $x_1$ | 0.8712 | 1 | 13.94 | 0.0073 | 11.62 | 1 | 99.25 | <0.0001 ** |
| $x_2$ | 0.0392 | 1 | 0.6272 | 0.4544 | 4.58 | 1 | 39.09 | 0.0004 * |
| $x_3$ | 0.0050 | 1 | 0.0800 | 0.7855 | 1.95 | 1 | 16.66 | 0.0047 |
| $x_1 x_2$ | 0.1260 | 1 | 2.02 | 0.1986 | 0.4290 | 1 | 3.67 | 0.0971 |
| $x_1 x_3$ | 0.0650 | 1 | 1.04 | 0.3417 | 2.03 | 1 | 17.35 | 0.0042 * |
| $x_2 x_3$ | 0.8930 | 1 | 14.29 | 0.0069* | 1.66 | 1 | 14.22 | 0.0070 * |
| $x_1^2$ | 0.9671 | 1 | 15.47 | 0.0056 | 9.11 | 1 | 77.85 | <0.0001 ** |
| $x_2^2$ | 0.7578 | 1 | 12.13 | 0.0102 | 3.40 | 1 | 29.04 | 0.0010 |
| $x_3^2$ | 1.29 | 1 | 20.70 | 0.0026 | 8.41 | 1 | 71.88 | <0.0001 ** |
| Residual | 0.4375 | 7 | | | 0.8193 | 7 | | |
| Lack of fit | 0.1950 | 3 | 1.07 | 0.4551 | 0.5734 | 3 | 3.11 | 0.1509 |
| Pure error | 0.2425 | 4 | | | 0.2459 | 4 | | |
| Total | 5.80 | 16 | | | 5.80 | 16 | | |

$p < 0.0001$ (highly significant **), $p < 0.05$ (significant *).

### 3.4.3. Influence of the Interaction of Various Factors on the Evaluation Index

Figure 21a is the response surface diagram of the influence of the interaction of the airflow inclination angle and dust remover angle on the impurity rate. It can be seen from the figure that under the interaction of two factors, when the dust remover angle is fixed, the impurity rate gradually increases with the increase of the airflow inclination angle, and the change range is obvious. When the airflow inclination angle is fixed, the impurity rate increases gradually with increasing dust remover angle, and the change range is also obvious. It can be seen from the response surface that the interaction of the airflow inclination angle and dust remover angle has a significant effect on the impurity rate, which is consistent with the results of the analysis of variance.

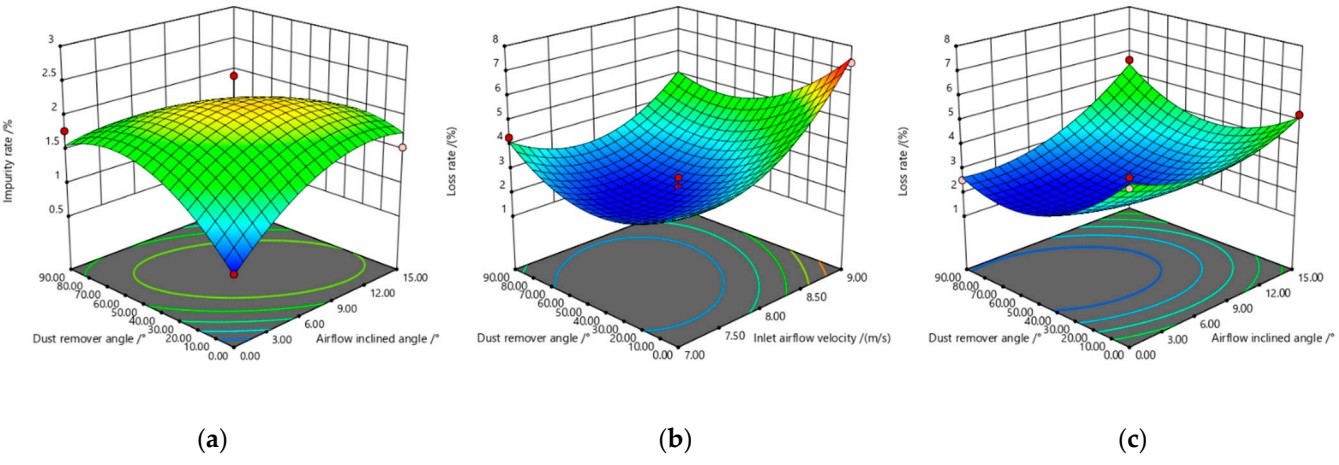

(a)                                  (b)                                  (c)

**Figure 21.** Influence of interactive factors on the impurity rate and loss rate.

Figure 21b is the response surface diagram of the influence of the interaction of inlet airflow velocity and dust remover angle on the loss rate. It can be seen from the figure that under the interaction of the two factors, when the dust remover angle is fixed, the loss rate gradually increases with the increase of the inlet airflow velocity, and the change range is obvious. When the inlet airflow velocity is fixed, the loss rate first decreases and then increases with increasing dust remover angle. It can be seen from the response surface that

the interaction of inlet airflow velocity and dust remover angle has a significant effect on the loss rate, which is consistent with the results of the analysis of variance.

Figure 21c is the response surface diagram of the influence of the interaction of the airflow inclination angle and dust remover angle on the loss rate. It can be seen from the figure that under the interaction of two factors, when the airflow inclination angle is fixed, the loss rate first increases and then decreases with the increase of the dust remover angle. When the dust remover angle is fixed, the loss rate first decreases and then increases with increasing airflow inclination angle. It can be seen from the response surface that the interaction of the airflow inclination angle and dust remover angle has a significant effect on the loss rate, which is consistent with the results of the analysis of variance.

### 3.4.4. Parameter Optimization and Test Verification

To further improve the cleaning efficiency of safflower petals, under various experimental factor level constraints, the minimum value of the rejection and loss rate was taken as the optimization index, and the full factor quadratic regression equation of the performance index was established to carry out an objective optimization consistent with the optimal working parameter determination:

$$\begin{cases} \min Y_1(X_1 X_2 X_3) \\ \min Y_2(X_1 X_2 X_3) \\ -1 \le X_1 \le 1 \\ -1 \le X_2 \le 1 \\ -1 \le X_3 \le 1 \end{cases} \tag{32}$$

The optimization module in Design-Expert data analysis software was used to optimize the regression models of the impurity rate and loss rate. Among them, the importance of the impurity rate is (+++++), and the importance of the cleaning loss rate is (++++). The optimum test indexes were obtained as follows: impurity rate 0.69% and loss rate 2.66%. The optimum combination of working parameters were as follows: inlet airflow velocity 7 m/s, airflow inclination angle 0°, and dust remover angle 25°.

To verify the accuracy of the optimized parameter model, the optimized parameters are used for test verification. The inlet airflow velocity, airflow inclination angle, and dust remover angle were set at 7 m·s$^{-1}$, 0° and 25°. A total of 5 groups of tests were conducted. Each group was tested twice, and the test results were averaged. The results of verification test are shown in Table 7.

**Table 7.** Results of verification test.

| Test No. | Impurity Rate/% | Loss Rate/% |
|---|---|---|
| 1 | 0.75 | 2.81 |
| 2 | 0.71 | 2.74 |
| 3 | 0.66 | 2.69 |
| 4 | 0.69 | 2.84 |
| 5 | 0.80 | 2.69 |
| Average value | 0.72 | 2.75 |
| Standard deviation | 0.03 | 0.09 |

It can be seen from the test results and particle size distribution that the impurity content is 0.72%, which is 0.03 percentage points lower than the predicted value. The loss rate is 2.75%, which is 0.09 percentage points lower than the predicted value. The comparison between theoretical optimization and experimental results shows that they are relatively close. The obtained optimal parameter combination meets the requirements of cleaning and can be used as the optimal parameter. The results of the test are shown in Figure 22.

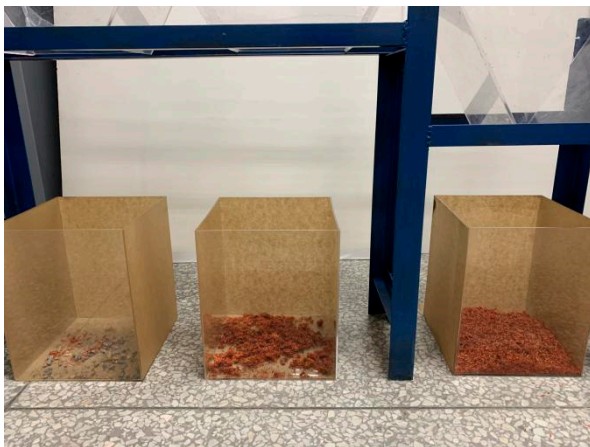

**Figure 22.** Experiment results.

### 4. Conclusions

In this paper, the cleaning efficiency of a safflower air-separation device under different inlet airflow velocities, airflow inclinations and dust remover angles was predicted by using an accurate shape model of the material particles and combining the advantages of computational fluid mechanics and the discrete element method. The stress status and movement tendency of safflower petals, bonded petals, and stones in the airflow field were analyzed. The conclusions are as follows:

1.  Due to the difference in the aerodynamic properties of safflower petals, bonded petals and stones, they have different movement trends in the airflow field. The smaller the mass of the material, the greater the axial velocity, the greater the flight distance and the greater the horizontal displacement, which resulted in various materials falling on different outlets. Due to the complex interaction force between particles of various shapes, the high-mass bonded petals and stones are carried to the safflower petals collecting part by quantities of low-mass safflower petals, thereby affecting the separation effect.
2.  The impurity rate and loss rate under different factors are compared with the simulation results by setting up the test, and the experimental results are found to be close to the simulation results. Due to the randomness of the physical shape and quality of the particles, the impurity and loss rates of the test are slightly larger than those of the simulation results.
3.  Combined with the Box–Behnken experimental design principle, three factors and a three-level response surface analysis method are used to optimize the parameters. With that minimum value of the impurity rate and the loss rate as the target, the optimal parameters obtained are as follows: the inlet airflow velocity was 7 m/s, the airflow inclination angle was 0°, and the dust remover angle was 25°. The verification test results show that the average impurity rate is 0.69%, and the average loss rate is 2.66%.

These results help us gain new insight and provide reliable methods for improving existing safflower air-separation devices.

**Author Contributions:** Conceptualization, Z.H. and H.Z.; software, Z.H; validation, Y.G., Z.H. and W.W.; resources, Z.H.; data curation, J.W.; writing—original draft preparation, Z.H.; writing—review and editing, Z.H. and H.Z.; project administration, H.Z.; funding acquisition, Y.G. All authors have read and agreed to the published version of the manuscript.

**Funding:** This research was funded by National Natural Science Foundation of China, grant number 52065057, and National Key R&D Program of China, grant number 2019YFC1710905.

**Institutional Review Board Statement:** Not applicable.

**Informed Consent Statement:** Not applicable.

**Data Availability Statement:** All relevant data presented in the article are stored according to institutional requirements and, as such, are not available online. However, all data used in this Manuscript can be made available upon request to the authors.

**Conflicts of Interest:** The authors declare no conflict of interest.

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
