# Peer review of "Simulation and Experiment of Gas-Solid Flow in a Safflower Sorting Device Based on the CFD-DEM Coupling Method"

_processes, doi:10.3390/pr9071239_

Round 1
Reviewer 1 Report
the most difficult in modeling of gas-solid flow is closing relation for multiphase flow. there are a lot of papers on the subject published during the last more than 60 years, therefore comparison of the multiphase models applied is very useful.
Reviewer 2 Report
The manuscript entitled “simulation and experiment of gas-solid flow in a safflower sorting device based on the CFD-DEM coupling method” is interesting and should be publishable in “Processes”. I have suggested major corrections as the CFD part is not well described and details are missing to make this work fully understood. The following points will need to be addressed before the manuscript can be accepted for publication:
Major points
- It is not clear what the authors mean by “… the mechanistic harvesting of safflower is less automated…” Less automated than what? Do the authors mean: “… the harvesting of safflower is not automated yet…”?
- “.. still artificial”. Not clear, what do the authors mean?
- There is no need to define Computational Fluid Dynamics (CFD) and Discrete element method (DEM) each time you mention them. They should be defined once, the first time they are written down, then you can use CFD and DEM everywhere directly.
- The references are listed differently in the text. For instance, on L67 where you have 3 different styles. Please use one everywhere in the text.
- Make sure that the caption/title is below/above the picture/table and not on another page (L119, L226, L254)
- “ … the gas phase is regarded as an incompressible fluid based on the Euler Lagrange method.” Not sure what this means. I believe the carrying flow is Eulerian and the particles are Lagrangian, but the gas phase is not based on the Euler-Lagrange method.
- It is not clear how you define the shape of the particles shown in Figure 5 in the DEM application. Please explain. Did you consider the real shapes shown in Figure 5, or did you consider spherical particles, with different properties?
- Use either “Eq.” or “Equation”, but not both.
- No information on the CFD, settings, DEM settings, schemes, time-step, convergence criterion, etc...
- How is the flow solver coupled with DEM. Details need to be provided regarding the software settings.
- Only the boundary conditions are mentioned, and the k-epsilon model mentioned (l256-258). What wall treatment did you consider, and why? What about the y+ value?
- The mesh independence study does not seem to have been done.
- Should it be Figure 9 instead?
- Should it be Figure 8 instead?
- Figure 8. I am not sure you should link the different points as it looks like you have a trend, but in fact, the velocity does not always increase on the x-axis, and all cases are very different.
- L300-302. “According… lower outlet” Not sure how you can see this from Figure 10... Besides, you have not defined what laminar/turbulent flows are.
- L311:”... it… starts… air flow” Not clear, to rewrite.
- Not clear so far how you defined bonded petals and stones in your model.
- L328, and L361. A bit strange to start Section 3.2.2 and 3.2.3 with 3 figures without text...
- L337 “We set… 45deg.” Why did you consider these conditions?
- L370: “We set... 0deg.” Why did you consider these conditions?
- L372-374. “For the convenience … 0.02”. Please remove, you have already made this statement in the previous section.
- L383: “excessively gentle” Not sure what this means…
- Section 3.3 Please replace “inclined angle” with “angle” or “inclination angle”
- L452 and 453. What are (+++++)?
- Line 469. Should it be Table 4 instead?
Minor points
- L22, please change “before” to “previous”
- L24: replace “to study” with “studying”
- Please remove “theoretical”
- L83: replace “by” with “through”
- L158: “As shown in..”
- Figure 3 is not well placed as it should be under or above the paragraph.
- L238, remove “is” -> “… the smaller the gravity, the larger…”
Round 2
Reviewer 2 Report
Many thanks to the author(s) for the fast action and for addressing all my previous comments and suggestions. I now only have very few minor revisions, mainly regarding English, see below. I would like to add that I really enjoyed reading this paper and I found the research and methods being used here very interesting.
Point 2: “.. still artificial”. Not clear, what do the authors mean?
Response 2: Thank you very much. I have changed it to “we are still harvesting safflower artificially rather than mechanically”.
(Line 40, Page 1 in the revised manuscript)
- Apologies for this, but I still don’t understand the meaning of “artificially” here. Could you explain how this is harvested right now? Is it done manually using some blowers of some sort?
- Please write EDEM in Line 232 rather than edem
- Please remove the following few sentences “Where laminar and turbulent flow is a property of fluid flow... property is called turbulent” on Lines 333-337.
